# The Role of Innate Immune Cells in Tumor Invasion and Metastasis

**DOI:** 10.3390/cancers13235885

**Published:** 2021-11-23

**Authors:** Yu-Kuan Huang, Rita A. Busuttil, Alex Boussioutas

**Affiliations:** 1Department of Medicine, Royal Melbourne Hospital, The University of Melbourne, Parkville, VIC 3010, Australia; tony.huang@unimelb.edu.au (Y.-K.H.); rita.busuttil@unimelb.edu.au (R.A.B.); 2Department of Gastroenterology, The Alfred Hospital, Melbourne, VIC 3004, Australia; 3Central Clinical School, Monash University, Melbourne, VIC 3004, Australia

**Keywords:** innate immunity, invasion, metastasis, cancer, innate immunology

## Abstract

**Simple Summary:**

Tumor invasion and metastasis are one of the main reasons patients succumb to cancer. In this review, we summarize recent studies which provide evidence on the involvement of cells of the innate immune system and their function in invasion and metastasis.

**Abstract:**

Metastasis is considered one of the hallmarks of cancer and enhanced tumor invasion and metastasis is significantly associated with cancer mortality. Metastasis occurs via a series of integrated processes involving tumor cells and the tumor microenvironment. The innate immune components of the microenvironment have been shown to engage with tumor cells and not only regulate their proliferation and survival, but also modulate the surrounding environment to enable cancer progression. In the era of immune therapies, it is critical to understand how different innate immune cell populations are involved in this process. This review summarizes recent literature describing the roles of innate immune cells during the tumor metastatic cascade.

## 1. Tumor Invasion and Metastasis

Metastasis is the general term used to describe the spread of cancer cells from the primary tumor to distant sites in the body and was reported by Hanahan and Weinberg as one of the hallmarks of cancer [1]. It has been reported that approximately 90% of cancer deaths are associated with metastases [2,3]. While metastasis has been considered to occur in later stages of disease progression, accumulation of earlier events during metastatic dissemination has been reported prior to clinically detectable tumors [4] and is one factor responsible for the concept of cancer with unknown primary.

The “metastatic cascade’’ describes the processes and restrictive bottlenecks through which tumor cells acquire cellular traits that allow them to exit primary tumor site and to travel and colonize distant organs. It involves major steps including local invasion, intravasation, and extravasation, which are well recognized in cancer [1,5,6] (Figure 1).

Local invasion occurs when tumor cells proliferate, accumulate genetic alterations, stimulate angiogenesis, and undergo epithelial-mesenchymal transition (EMT) locally. This result in the loss of cell-cell adhesion, degradation of the extracellular matrix (ECM), and basement membrane and allows for increased motility of the tumor cells to invade the adjacent stroma and is an important trait of malignancy [7,8].

In addition to the reliance on the blood supply of their host, tumor cells secrete angiogenic factors inducing abnormal vasculatures that are often leaky and immature [9]. These features result in tumor hypoxia which drives EMT of tumor cells and provide a route for mesenchymal cells to enter blood circulation; hence, there is a process of intravasation [9,10]. Lymphatic intravasation is an alternative pathway by which tumor cells can enter the circulation and metastasis to regional and distant lymph nodes [11].

The following step of the cascade is successful extravasation [5]. This requires tumor cells (A) to survive and exit the circulatory system, (B) to escape the immune system and persist in a relatively silent state while retaining a stem-like tumor initiating capacity, until (C) they adapt to local environment by reaching a supportive niche, and finally (D) establish a clinically detectable metastatic disease [5]. The majority of disseminated cells from primary tumors are eliminated by the immune system in circulation and in distant organs or remain dormant at metastatic sites and only very few develop into metastatic tumors [12,13]. Successful establishment of these metastatic tumors require tumor cells to metastasize to a favorable local microenvironment, as described by the “seed and soil” hypothesis [14].

## 2. Metastasis Modulated by Tumor Microenvironment and Immunity

While tumor cell-intrinsic factors including tumor heterogeneity are known to drive invasion and metastasis, extrinsic factors within the tumor microenvironment (TME) have been shown to play critical roles in determining the fate of cancer progression [15]. The stromal components of the TME are composed of several cell populations, including immune cells, fibroblasts, and endothelial cells [16]. The interaction between these cell types and the ECM, vascular distribution, oxygen (hypoxia), nutrients, and chemokines further complicates the tumor milieu [17].

The immune system has been demonstrated to shape the local TME [18] and contributes not only to the elimination of tumor cells, but may also be subverted to support their invasion and metastasis [19,20]. Under normal conditions, the Cancer-Immune cycle involving the dynamic communication of both innate and adaptive immune systems is activated to remove malignant cells in a process known as immune surveillance [21]. This cycle initiates as neo-antigens are taken up by antigen-presenting cells (APCs) of the innate immune system and presented to the adaptive immune system to acquire specific recognition of tumor cells. These activated T cells then migrate to the tumor site allowing them to target the tumor cells expressing these neo-antigens. Effector molecules such as interferon gamma (IFN-γ), tumor-necrosis factor alpha (TNF-α), as well as recognizing molecules (e.g., NKG2D) have been shown to be important in this process [22,23]. Functional immune surveillance results in the elimination of tumor cells; however, this is often disrupted or hijacked by cancer cells at multiple different points resulting in immune escape [21,24]. Possible routes include, but are not limited to, (A) downregulation or impairment of antigen presentation ability [25], (B) recruitment of pro-tumorigenic immune cells and secretion of inflammation-regulating cytokines by cells in the TME [26], and (C) upregulation of the expression of immune inhibitory molecules to suppress immune cell cytotoxicity [27]. Approaches that aim to restore an effective Cancer-Immune cycle are being investigated and are referred to as immune checkpoints [21].

Innate immunity (e.g., dendritic cells, macrophages, neutrophils, and natural killer cells) represents the first line of defense and rapid response to foreign substances and in anti-tumor activities [28]. However, these immune cells have also been shown to drive cancer-associated inflammation [1] and to participate in several steps of the metastatic cascade from reorganizing ECM [29], modulating vessel formation and permeability [9], and influencing cancer cell motility and states [30,31] (Figure 2). This remainder of this review summarizes known characteristics of innate immune cell populations on tumor invasion and metastasis and discusses recent advances in the field.

## 3. Dendritic Cells, Macrophages, Neutrophils, and MDSCs

### 3.1. Dendritic Cells

Dendritic cells (DCs) are professional APCs that bridge the innate and adaptive immune systems and are essential to a competent Cancer-Immune cycle [32]. Different DC subsets include plasmacytoid DC (pDC), which are major producers of type-I interferons (IFNs), and conventional DC (cDC), which specialize in presenting antigen to naïve T cells [33]. cDCs can be further divided into cDC1 and cDC2 based on their surface phenotype and lineages [34].

Interrupted DC development and dampened antigen-presentation in DCs allows tumor cells to evade immune recognition [35]. Examples include the observation of breast and pancreatic tumor-derived granulocyte colony-stimulating factor (G-CSF) inhibiting cDC1 development by impairing interferon response factor 8 (IRF8) expression in DC progenitor cells [36]. A deficiency of basic leucine zipper ATF-like transcription factor 3 (BATF3), a transcription factor required for cDC differentiation and programming CD8+ T cell memory [37,38], resulted in a defective DC-associated, cytotoxic T cell-mediated anti-tumor immunity [39] and natural killer (NK) cell-suppressed metastasis in vivo [40]. CD8α+/XCR1+ cDC1 was found to fail in cross-presenting to CD8+ T cells and tumor rejection in the absence of WDFY4 gene, but these mice had normal lymphoid and non-lymphoid cDC1 populations compared to BATF3-deficient mice [41].

In addition, tumor-driven immune-suppressing DC subsets have been described in the context of tumorigenesis and metastasis [42]. Immune suppressive pDCs were found to support melanoma progression by promoting regulatory immunity through OX40L and inducible T cell co-stimulator ligand (ICOSL) [43]. In ovarian cancer, tumor-driven unremitting expression of Satb1 was reported to enhance Galectin-1 and interleukin 6 (IL-6) production in activated inflammatory DCs, resulting in immune suppression and contributing to malignant progression [44]. Tumor cell-secreted prostaglandin E2 (PGE2) and transforming growth factor β (TGFβ) promoted the immunosuppressive activity of immunocompetent DCs with upregulation of programmed death-ligand 1 (PD-L1) [45]. In an in vivo model of pancreatic ductal adenocarcinoma, early liver metastases were associated with MGL2 and PD-L2-expressing DCs derived from tumor-released granulocyte macrophage colony-stimulating factor (GM-CSF) [46]. Preferential expansion of T regulatory cells by these DCs were observed in vitro and in vivo and selectively targeting of this DC population with in Mgl2-depletion or PD-L2-blockage resulted in activated CD8 T cell and suppressed tumor metastasis [46]. Finally, DCs have also been reported to induce a premetastatic niche during lymph node cancer dissemination, via cyclooxygenase-2 (COX-2)/EP3-dependent induction of stromal cell-derived factor 1 (SDF-1) in a murine model of Lewis lung carcinoma [47].

### 3.2. Macrophages

Macrophages were first described as the “big (macro-) eater (-phage)” and are now recognized by their multi-functionality and plasticity [48]. Fully polarized M1 (classically activated) and M2 (alternatively activated) macrophages represent the extreme ends of a spectrum of functional states [49].

The M1 phenotype is induced by IFN-γ, lipopolysaccharide (LPS), and TNF-α, and is associated with the enhanced production of pro-inflammatory cytokines and chemokines [50,51], as well as antigen presentation/co-stimulatory molecules [52,53,54]. M1 macrophages function through inducible nitric oxide synthase (iNOS) [48] imparting a direct anti-tumor effect of nitric oxide via the inhibition of mitochondrial electron transport [55,56].

The M2 phenotype is polarized by IL-4 or IL-13 [57]. These macrophages produce high levels of angiogenic and ECM remodeling factors, including vascular endothelial growth factor (VEGF), TGFβ, matrix metalloproteinases (MMPs), and arginase-I (ARG1), and exhibit enhanced phagocytosis of apoptotic cells [49,58,59]. Macrophage colony-stimulating factor (M-CSF) is critical for the development of these macrophages [60], as well as the enhancement of macrophage-mediated extravasation of tumor cell [61]. In summary, M2 macrophages have been described as experts in resolving inflammation, promoting wound healing, and suppressing active anti-cancer immunity [62].

Due to the complexity of activation status observed in macrophage populations that extended beyond the simplified M1/M2 system, further in vitro-polarized subpopulations have been described to distinguish different functional populations [63]. More recently, a multi-dimensional spectrum model of macrophage activation/polarization was proposed using non-supervised clustering methods to examine the transcriptomic signatures of in vitro-polarized human macrophages treated with a combination of stimuli [64].

Tumor-associated macrophages (TAMs) are associated with tumor progression and metastasis [65] and substantial TAM heterogeneity has been reported in different cancer models [66,67,68,69]. TAMs of hematopoietic origin are recruited locally to the TME from peripheral blood monocytes and are polarized into TAMs via tumor-derived M-CSF, CCL2, and SDF-1 [70]. In contrast, TAMs may also originate from tissue-resident macrophages (TRM). These are a population of macrophages derived from embryonic precursors and seeded in different organs from birth. TRMs are responsible for maintaining tissue homeostasis and have been proposed to be less plastic in nature compared to monocyte-derived macrophages [71,72]. In a mouse pancreatic ductal adenocarcinoma model, TAMs of hematopoietic stem cell origin demonstrated an enhanced immune activation antigen presentation ability, whereas tissue-resident TAMs were found to proliferate in situ and exhibited an ECM remodeling phenotype [73]

Within the TME, TAMs have been shown to migrate deep into tumor hypoxic regions where they can support tumor cell migration and invasion via ECM remodeling by the upregulation of MMP-2, -7, and -9, and stimulate angiogenesis through enhanced VEGFA production [74,75]. Tumor hypoxia was reported to facilitate the forming and intravasation of highly metastatic circulating tumor cell clusters [76], and TAMs were shown to sense and respond to lactate signal from tumor cells within hypoxia through G protein-coupled receptor 132 (Gpr132) and stimulate cancer metastasis to distant organs [77]. Using an in vivo mammary tumor model, a synergistic interaction related to epidermal growth factor (EGF) and M-CSF stimulation between macrophages and tumor cells during cell migration and intravasation was observed [78]. VEGFA signaling from TIE2-expressing macrophages induced local loss of vascular junctions and transient vascular permeability, stimulating tumor cell intravasation [79]. In addition, CXCR4+ TAMs were found to migrate toward CXCL12-expressing perivascular cancer-associated fibroblasts, and once located on the blood vessel, TAMs became sessile and promoted tumor cell intravasation [80].

While macrophages have been shown to recognize and eliminate circulating tumor cells [81], there is increasing evidence of their roles in establishing pre-metastatic environments. In a mouse model of pancreatic cancer, liver-resident macrophages that uptake tumor-derived exosomes containing high macrophage migration inhibitory factor (MIF) showed an elevated TGFβ secretion [82]. This in turn upregulated fibronectin production by hepatic stellate cells, creating a fibrotic microenvironment suitable for liver pre-metastatic formation [82]. Similarly, CD206+/TIE2+ macrophages recruited by tumor-derived CCL2 induced upregulated Wnt-1 and downregulated E-cadherin in early HER2+ tumor cells, establishing a pre-metastatic niche promoting early cancer dissemination and intravasation [83]. Macrophage-expressed integrin α4β1 has been reported to interact with tumor cell-expressed vascular cell adhesion molecule 1 (VCAM-1), resulting in sustained survival of these newly extravasated tumor cells and elevated local macrophage activity within distant metastatic sites [84,85].

### 3.3. Neutrophils

Neutrophils are polymorphonuclear phagocytes which play a role in the front line of host immune response against pathogens and effector cells during inflammation [86]. In different cancers, an increased number of tumor-associated neutrophils (TANs) and high neutrophil/lymphocyte ratio have been associated with poor patient prognosis [87,88,89] and metastatic relapse [90,91,92]. Similar to the M1/M2 model of TAMs, it has been proposed that TANs exist in N1/N2 polarization states to describe anti- and pro-tumor populations, respectively [93]. However, it is also recognized that the heterogeneity of functional TANs is beyond the scope of this bipolar model [94].

TANs have been shown to contribute to tumor angiogenesis and intravasation via the secretion of MMP-9 [95] and counteract anti-VEGF therapy in metastatic colorectal cancer [96]. G-CSF, an essential glycoprotein which stimulates neutrophil survival and differentiation, was found to mobilize neutrophils in promoting lung metastases of breast tumor cells [97]. Inflamed neutrophils have been shown to promote the rate of tumor cell extravasation in a zebrafish model [98], pre-establish lung microenvironment for metastatic breast tumor cell colonization [99], and facilitate Lewis lung carcinoma cell metastasis through neutrophil protease-mediated degradation of thrombospondin-1 in mice [100]. TANs were reported to suppress intraluminal NK cell-mediated tumor cell clearance and enhance extravasation of disseminated carcinoma cells [101] and work in conjunction with gamma-delta T cells to promote breast cancer metastasis in vivo [102].

While TANs have been reported as anti-tumor through reactive oxygen species (ROSs) or antibody-dependent cellular cytotoxicity (ADCC) [103], the influence of TANs on tumor metastasis may be relevant to other immune components within the TME [104]. It was demonstrated that in a murine model of breast cancer, G-CSF-expanded neutrophils were inhibitory of lung metastatic colonization in NK cell-deficient mice. In contrast, using the same tumor models in NK cell-competent mice, neutrophils facilitated metastatic colonization by suppressing the anti-tumor NK cells while remaining tumoricidal themselves [104].

Neutrophil-derived elastase was associated with the promotion of tumor inflammation and acceleration of lung cancer cell proliferation and invasion [105,106] through elastase-mediated formation of dilated intratumoral vasculature [107].

Neutrophil extracellular traps (NETs) are neutrophil-generated extracellular fibers consisting of granule proteins and chromatin that immobilize pathogens to facilitate their subsequent elimination [108]. In a cancer setting, serum NET was predictive of the occurrence of liver metastases in patients with early-stage breast cancer [109]. Local NET expression has been shown to induce resistance to radiation therapy in invasive bladder cancer patients [110], to be positively associated with progression of liver metastases in colorectal cancer patients, and to accelerate metastatic disease in a murine model post-surgery [111]. The DNA component of NETs was found to act as a chemotactic factor to attract tumor cells through an extracellular DNA-sensor, CCDC25, expressed on tumor cells which subsequently activated the integrin-linked kinase (ILK)-β-parvin pathway to enhance cell motility [109]. Additionally, NET was found to sequester circulating tumor cells to increase cell adhesion and promote hepatic and lung metastases [112,113], and to awaken dormant tumor cells at distant metastatic sites through ECM remodeling and sustained local inflammation [114].

### 3.4. Myeloid-Derived Suppressor Cells

Myeloid-derived suppressor cells (MDSC) are a diverse of population of immature myeloid cells found in the TME that have potent immune suppressive activity [115] and capacity to establish a tumor pre-metastatic niche [116]. Within the heterogeneous MDSC, the two main functionally discrete populations, granulocytic (G-MDSC) and monocytic (M-MDSC), have been described [116].

G-MDSCs have been reported to suppress T cell responses through ROS inducing antigen-specific tolerance, while M-MDSCs showed stronger suppressive activity in both antigen-specific and non-specific responses via enhancing iNOS, arginase-I (ARG1) and other immune suppressive cytokines [117]. Importantly, these two populations were found to contribute to different phases of the metastatic cascade in a murine breast cancer model [118]. Pulmonary G-MDSCs were reported to support the metastatic establishment by reverting the tumor EMT phenotype and promoting their proliferation, whereas the tumor-infiltrated M-MDSCs facilitated malignant cell dissemination from the primary tumor site by inducing EMT and the cancer stem cell phenotype [118].

## 4. Natural Killer Cells and Other Innate Lymphoid Cells

Innate lymphoid cells (ILCs) are predominantly tissue-resident lymphocytes that, unlike T and B cells, do not express diversified antigen receptors [119]. ILCs have been shown to regulate tissue immunity, inflammation, and homeostasis [120] and can be classified into five groups, namely (conventional) natural killer (NK) cells, group 1 ILCs (ILC1s), ILC2s, ILC3s, and lymphoid-tissue inducer cells (LTis) [121].

### 4.1. Natural Killer Cells

NK cells express a wide range of inhibitory and stimulatory receptors which play a role in immune surveillance in cancer [122]. Increased density of intratumoral NK cells has been associated with favorable patient overall survival [123], and activated NK cells were observed in patients with prolonged metastasis-free disease [124]. Reduced NK cell cytotoxic activity and increased peripheral blood circulating tumor cell numbers were observed in patients with metastatic breast, prostate, and colorectal cancer [125].

Functionally, tumor cells downregulating major histocompatibility class I (MHC-I) molecules can be recognized by NK cells resulting in programmed cell death [122]. Natural cytotoxicity triggering receptor 1 (NCR1 or NKp46)-mediated IFN-γ produced by NK cells have been shown to increase ECM protein fibronectin-1 resulting in decreased metastasis formation in both human and mice [126]. NK cell-mediated tumor cell recognition and cytotoxicity through NCRs and DNAX accessory molecule-1 (DNAM-1) was observed in lymph node metastases-derived human melanoma cell lines [127]. Deletion of myeloid cell leukemia sequence-1 (*Mcl1*) and heparinase in NK cells resulted in the outgrowth of metastatic diseases in mice [128,129].

The TME can alter NK cell phenotype. Tissue-intrinsic NK cell suppression was observed in prostate, which was mediated partly by high local TGFβ secretion and can be enhanced upon tumor cell infiltration [124]. When exposed to TGFβ, upregulation of VEGF and placental growth factor (PIGF) was observed in peripheral blood NK cells from healthy subjects, and a similar pro-angiogenic phenotype was observed in surgically resected non-small cell lung cancer patient samples [130]. Suppression of NK cell-mediated immunosurveillance via the Smad3-E4BP4 signaling pathway contributes to tumor growth, invasion, and metastasis in syngeneic mouse tumor models [131]. Tumor-derived metabolites, hypoxic environment, and low pH were shown to impair NK cell activity [132], inducing NK cell apoptosis and promoting tumor metastasis [133,134].

### 4.2. Innate Lymphoid Cells

ILCs belong to lymphoid lineage but do not express antigen specific receptors. They reside primarily at mucosal surfaces and in healthy individuals within a symbiotic relationship with resident microflora to induce an appropriate immune response [119].

ILC1s exhibit a production of inflammatory cytokines and show gene-expression patterns overlapping with conventional NK cells [121,135]. Unlike conventional NK cells, subsets of tissue-resident ILC1s have been identified in small intestinal mucosa, liver, and salivary glands, where its population was maintained predominantly via local self-renewal rather than replenishment from blood [136]. Increased percentages of ILC1s in peripheral blood mononuclear cells (PBMCs) that exhibited reduced TNF-α production have been observed in acute myeloid and chronic lymphocytic leukemia patients [137] and in metastatic colorectal patient post-chemotherapy [138]. In contrast, tissue resident ILC1s and conventional NK cells were shown to contribute to controlling metastatic seeding in liver and restraining tumor outgrowth in mice, respectively [139]. Importantly, the killing capacity of these ILC1s was independent of the metastatic microenvironment, compared to a cancer type-dependent fashion in conventional NK cells [139].

ILC2s are mucosal tissue resident cells that are characterized by Th2 cytokine production (e.g., IL-4 and IL-13) and contribute to type 2 inflammation [121,140]. A previous study identified a subset of IL-5 producing ILC2s, which was able to regulate the number of eosinophils in the lung to prevent the development of metastasis [141]. IL-33-dependent tumor-infiltrating ILC2s have been demonstrated to mediate tumor immune surveillance in cooperation with DCs in promoting adaptive cytolytic T cell responses and controlling tumor metastasis in mice [142]. In an acute promyelocytic leukemia model, elevated tumor-derived PGD2 and NKp30-B7H6 engagement was found to activate ILC2s, which in turn drove the activation of M-MDSCs via IL-13 secretion [143]. An increased percentage of IL-13-expressing intratumoral ILC2 was observed in parallel with the recruitment of immune-suppressive TAMs, IL-10-expressing DCs and MDSCs in an IL-33-treated breast cancer model with accelerated tumor progression and lung/liver metastases [144]. Activation of lung-resident IL-33-dependent ILC2 drove IL-5-induced eosinophilia, resulting in a profound local suppression of IFN-γ production and cytotoxic function of anti-metastatic NK cells [145].

ILC3s are a heterogeneous population but are RORγt-dependent for their development and function [146]. ILC3s represent a source of IL-22 as well as IL-17, GM-CSF, and IFN-γ and contribute to inflammatory diseases [121,147]. The increased density of ILC3s was reported to be associated with increased lymphatic tumor cell invasion and metastasis in human breast cancer patients [148] and with a subset of NCR-expressing ILC3s, which were proposed to contribute to intratumoral lymphoid structures in human lung cancer [149]. These NCR+ILC3s were mainly localized at the edge of tumor-associated tertiary lymphoid structures, to produce IL-22, TNF-a, IL-8, and IL-2, and when activated, are able to activate endothelial cells [149].

IL-12-stimulated ILC3s have been shown to control primary melanoma growth in mice; however, their tumor-suppressive function was dependent on different tissue microenvironments [150]. Rapid intestinal tumorigenesis in vivo was observed to be induced by resident IL-23R+ ILC3s prior to the recruitment of inflammatory infiltrates [151]. IL-22 produced by colonic ILC3s was identified to mediate the development of invasive colon cancer by selectively acting on epithelial cells to induce STAT3 phosphorylation and cell proliferation [152]. Finally, CCL21-mediated recruitment of ILC3 to breast tumors in mice stimulated the stromal cell production of CXCL13 and facilitated tumor lymphatic vessel invasion [148].

## 5. Other Cell Types Associated with Innate Immunity

### 5.1. Mast Cells

Mast cells (MCs) are tissue-resident, multifunctional effector cells which play a crucial role in allergic responses and infectious diseases [153]. MCs respond to extrinsic signals by secreting prestored histamine and proteases, newly synthesized inflammatory mediators, and express high-affinity receptors for immunoglobulin E (IgE) [153,154,155].

In cancers, increased MC density was correlated with poor prognosis and increased metastasis in multiple tumor types [156,157]. MCs have been reported to accumulate at the tumor margins as well as peri-vascular regions and were associated with the increased blood vessel density in the TME [158,159]. Tumor-derived adrenomedullin (ADM) induced degranulation in MCs via the PI3K-AKT signaling pathway, which effectively promoted proliferation and inhibited apoptosis of gastric tumor cells dependent on MC-produced IL-17A [160]. MCs activated by tumor-derived IL-33 have been reported to promote gastric cancer progression through macrophage mobilization [161] and MC-derived chymase has been shown to promote lung tumor cell EMT phenotype in vitro [162]. Exosomes derived from lung tumor cell lines was found to activate MCs through stem cell factor (SCF) signal transduction, which led to MC degranulation and the release of tryptase, resulting in accelerated tumor proliferation and invasion [163].

### 5.2. Basophils

Basophils represent <1% of human peripheral leukocytes and are known producers of IL-3, IL-4, IL-13, IgE, VEGFs, and other pro-angiogenic molecules [164]. Hepatocyte growth factor (HGF) is an angiogenic and autocrine growth regulator in chronic myeloid leukemia, which basophils were identified to be a major source in these patients [165]. An increased number of basophils has been shown to occur in advanced chronic and acute myeloid leukemia [166,167] and decreased basophils were associated with poor prognosis and enhanced venous/perineural invasion in colorectal cancer patients [168]. An enrichment of IL-4-expressing basophils was observed in tumor-draining lymph nodes of pancreatic ductal adenocarcinomas patients and investigation in relevant mouse models identified a monocyte and T cell partial-dependent recruitment of tumor-promoting basophils by CCL7/MCP3 and IL-3, respectively [169]. Finally, activated human and mouse basophils exhibit the ability to produce extracellular DNA traps as neutrophils [170,171], suggestive of their potential roles in promoting metastasis.

### 5.3. Eosinophils

Eosinophils are associated with protection against helminths, viral and bacterial pathogens and with the production of pro-inflammatory and pro-angiogenic factors [172]. Increased infiltration of eosinophils has shown tumor-dependent effects on patient survival [173,174,175,176]. Human eosinophils were reported to exert TNF-α and granzyme A-mediated tumoricidal activity toward colon carcinoma cells, to induce tumor cell apoptosis and to promote cell-cell adhesion via IL-18 pathway [177,178]. Lung recruitment of eosinophils was observed post administration of IL-33 in mice, which prevented the onset of pulmonary metastasis and suppressed primary melanoma tumor growth [179]. Recruitment of eosinophils via IL-25 and IL-33 by innate IL-5-producing cells was shown to be critical in suppressing tumor cell dissemination to the lung [141]. However, administration of IL-5 in mice facilitated lung metastatic colonization associated with eosinophilic inflammation [180]. Mechanistically, eosinophils were found to promote tumor cell migration and initiate metastasis through CCL6 signaling, which can be suppressed through CCR1 inhibition [181].

## 6. Summary

Tumor invasion and metastasis remain a major cause of cancer mortality. The engagement of innate immune cells with tumor cells is present at each stage of the metastatic cascade, which contributes to (A) regulating the survival and proliferation of tumor cells and (B) modulating and establishing an environment suitable for local invasion and distant metastasis (Figure 2, Table 1).

Tumor cells and/or the TME have been shown to dampen DC-mediated tumor cell recognition and NK cell/eosinophil-mediated anti-tumor activity. TAMs, TANs, and MSDCs were found to not only assist in ECM remodeling, tumor angiogenesis, and immune suppression, but also in the forming of pre-metastatic niche and supporting circulating tumor cells survival. Mast cells, basophils, and ILCs were involved in modulating anti-tumor immunity, EMT of tumor cells, and shaping of the tumor vasculature and lymphatic system. These significant inroads into understanding the roles of innate immunity during cancer progression provide routes for potential therapeutic interventions.

## Figures and Tables

**Figure 1 cancers-13-05885-f001:**
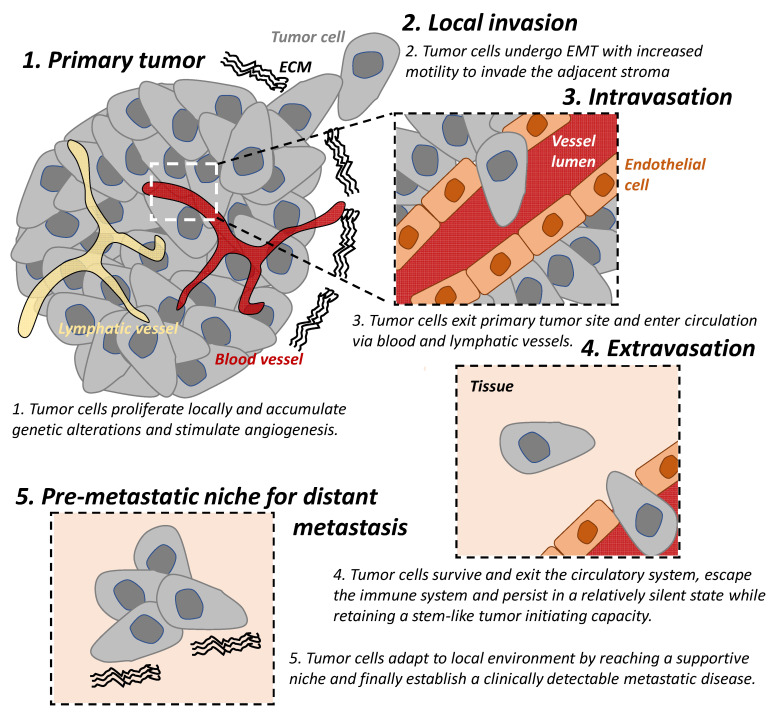
The tumor metastatic cascade. The “metastatic cascade” describes the processes and restrictive bottlenecks through which tumor cells acquire cellular traits, allowing them to exit the primary tumor site and migrate to distant organs which they then colonize. This occurs via a series of steps including local invasion, intravasation, and extravasation. EMT: epithelial-mesenchymal transition. ECM: extracellular matrix.

**Figure 2 cancers-13-05885-f002:**
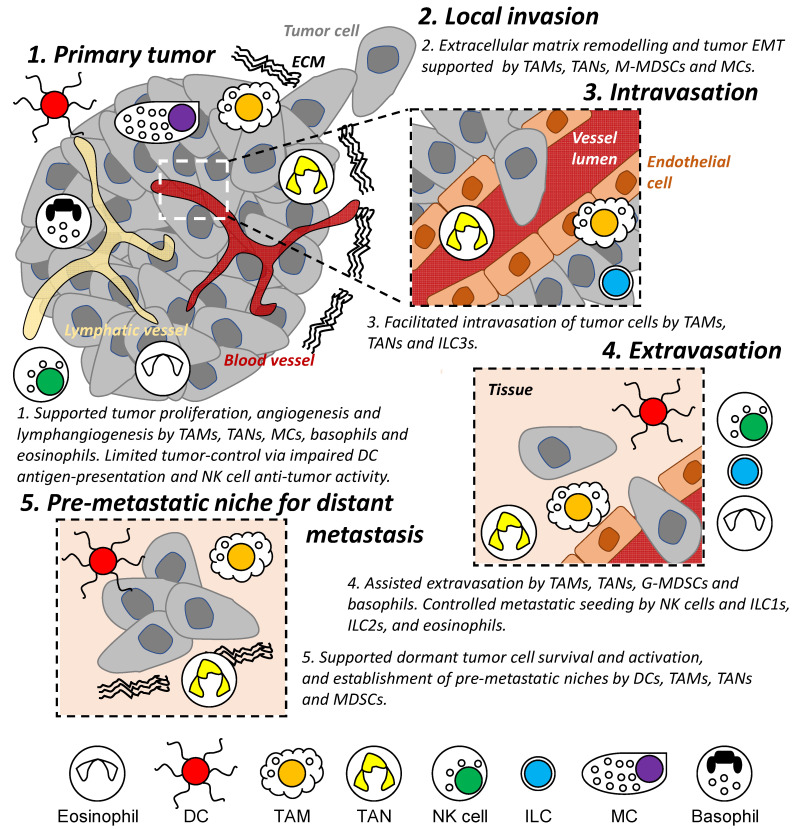
Role of innate immune cells during the metastatic cascade. Different populations of innate immune cells are involved in different stages of the metastatic cascade both locally and in distant organs/tissues, including regulating the survival and proliferation of tumor cells and assisting in the establishment of a permissive tissue environment enabling tumor cell invasion and migration. DC: dendritic cell. TAM: tumor-associated macrophage. TAN: tumor-associated neutrophil. M- and G-MDSC: monocytic and granulocytic-myeloid-derived suppressor cells. MC: mast cell. NK: natural killer. ILC: innate lymphoid cell. EMT: epithelial-mesenchymal transition. ECM: extracellular matrix.

**Table 1 cancers-13-05885-t001:** Role of innate immune cells during the metastatic cascade.

Cell Type	Potential Roles		Ref.
	In Cancer	In Metastatic Cascade	
Dendritic cells	antigen-presentation, immune regulation	local invasion, distant metastasis	[35,41,42,45,47]
Macrophages	pro- and anti-inflammation, immune regulation, pro- and anti-tumor, pro-angiogenic, EMT	local invasion, intravasation, extravasation, distant metastasis	[73,76,77,79,84,85]
Neutrophils	pro- and anti-inflammation, immune regulation, pro- and anti-tumor, pro-angiogenic	local invasion, intravasation, extravasation, distant metastasis	[95,99,100,105,106,114]
Myeloid-derived suppressor cells	immune regulation, pro-tumor, EMT	local invasion, distant metastasis	[117,118]
Natural killer cells	pro-angiogenic, anti-tumor	local invasion, extravasation, distant metastasis	[130,133,134]
Innate lymphoid cells	pro-inflammation, immune regulation, pro- and anti-tumor	local invasion, intravasation, extravasation, distant metastasis	[139,145,149,152]
Mast cells	pro-angiogenic, pro-tumor, EMT	local invasion	[158,159,162,163]
Basophils	pro-angiogenic, pro-tumor	local invasion, extravasation	[164,169,170,171]
Eosinophils	pro-inflammation, anti-tumor	local invasion, extravasation, distant metastasis	[177,178,179,181]
EMT: epithelial-mesenchymal transition.			

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
