# Peer review of "The Role of Innate Immune Cells in Tumor Invasion and Metastasis"

_cancers, 2021, doi:10.3390/cancers13235885_

Round 1

Reviewer 1 Report

The Authors don't apport the relevant scientific contribute.

Author Response

Response to reviewer’s comments

Reviewer 1:

Comments and Suggestions for Authors:

The Authors don't apport the relevant scientific contribute.

Whilst the clinical advantage of immunotherapy for cancer primarily focused on targeting adaptive immune system, there is increasing evidence that the innate immune cells are involved in either promoting anti-tumour environment or recruiting adaptive immune cells. However, it is also known that majority of the innate immune cells have pro-tumour or immune suppressive characteristic which resulted in their associations with tumour progression.

There are numerous reviews and meta-analyses already published on how each of the innate immune cell population was related to tumour invasion and metastasis, however, many which included the studies that were related to:

  1. Reduced density of anti-tumour immune cells à tumour invasion and metastasis
  2. Reduced anti-tumour activity / Increased immune suppression à tumour invasion and metastasis

In our review, we have focused specifically on recent studies that provided evidence on the involvement of innate immune cells during each step of the metastatic cascade, not just correlations. This will provide further accessibility to future studies focusing on limiting tumour invasion and metastasis, which are one of the main reasons why patients succumb to the disease.

We believe that we are providing some scientific contribution to the field.

Reviewer 2 Report

The manuscript by Huang et al describes the role of innate immune cells in tumor invasion and metastasis in general. Considering metastasis is one of the Hallmarks of Cancer and enhanced tumor invasion and metastasis is significantly associated with majority of the cancer-related mortality, this review is important to understand the cellular contributions other than the tumor epithelial cells. Thi review described one third of the cellular component of the tumor microenvironment. As rightly pointed out the innate immune components of the microenvironment have been shown to engage with tumor cells and not only regulate their proliferation and survival, but also modulate the surrounding environment to enable cancer progression. This review summarizes the roles of innate immune cells during cancer invasion and metastasis.

Though the review is interesting and simple to understood by any researcher without much background in immunology, it needs to lay out the functional interplay of this cell types for proper understanding. A little description and a schematic would help to address this.

Some minor comments

  1. In addition to the reliance on the blood supply of their host, tumors make (abnormal vasculatures their own vasculatures) that are often leaky, immature and morphologically abnormal [9)- the sentence can be rewritten as in most cases, the vasculature is induced by the tumor cells by secreting more and more angiogenic inducing factors due to the compendium of factors.

Author Response

Response to reviewer’s comments

Reviewer 2:

Comments and Suggestions for Authors

The manuscript by Huang et al describes the role of innate immune cells in tumor invasion and metastasis in general. Considering metastasis is one of the Hallmarks of Cancer and enhanced tumor invasion and metastasis is significantly associated with majority of the cancer-related mortality, this review is important to understand the cellular contributions other than the tumor epithelial cells. This review described one third of the cellular component of the tumor microenvironment. As rightly pointed out the innate immune components of the microenvironment have been shown to engage with tumor cells and not only regulate their proliferation and survival, but also modulate the surrounding environment to enable cancer progression. This review summarizes the roles of innate immune cells during cancer invasion and metastasis.

Though the review is interesting and simple to understood by any researcher without much background in immunology, it needs to lay out the functional interplay of this cell types for proper understanding. A little description and a schematic would help to address this.

Figure 2 highlights the location and proposed interplay of different cell types during the metastatic cascade. The majority of published literature focuses on specific cell types and lacks information of the changes of other cell types in the same environment. We therefore lack evidence to conclusively state that different cell types are interacting even though they may be involved in the same processes.

Some minor comments

  1. In addition to the reliance on the blood supply of their host, tumors make (abnormal vasculatures their own vasculatures) that are often leaky, immature and morphologically abnormal [9)- the sentence can be rewritten as in most cases, the vasculature is induced by the tumor cells by secreting more and more angiogenic inducing factors due to the compendium of factors.

We have amended this to

In addition to the reliance on the blood supply of their host, tumor cells-secrete angiogenic factors inducing abnormal vasculatures that are often leaky and immature. 

Reviewer 3 Report

In this work, Huang et al., review the effects of different populations of innate immune cells on antitumor targets, in order to be able to provide possible therapeutic interventions, focusing especially on the metastasis stage of a tumour process. In the first place, they explain the consequence of tumour invasion and the appearance of metastasis, with an emphasis on metastasis modulated from extrinsic factors of the tumour environment and how the first line of defence could promote cancer-associated inflammation or participate in the metastatic cascade. Subsequently, they describe the different types of cells, grouping into three groups according to the authors’ criteria.

The authors make an extensive review of a topic already developed and with multiple studies at different levels. However, the metastasis is a topic in full development at the present. Many recent and important references are used, the topics are well arranged and a very scientific and clear writing of language is used. All these things facilitate the understanding and make it easy to read. The quality of the figure is good.

Minor comments:

- Page 2, chapter 2: The authors describe different effects by which the immune system contributes to the removal of tumour cells or support the invasion and metastasis of other organs. The authors should describe more deeply the effect of cytokines secreted by immune cells on tumour progression.  Is there any reference in the literature on how the abundance of certain cells from TME influences the prognosis of cancer?

- Page 5: It would be interesting if the authors could broaden the topic of the relationship between neutrophils-MDSCs-NK cells

- Page 7: The immunosuppressive and protective properties of ILC3 and its contribution to the development of TLSs in lung cancer should be describe more deeply as it is reported in the reference 148.

Author Response

Response to reviewer’s comments

Reviewer 3:

Comments and Suggestions for Authors

In this work, Huang et al., review the effects of different populations of innate immune cells on antitumor targets, in order to be able to provide possible therapeutic interventions, focusing especially on the metastasis stage of a tumour process. In the first place, they explain the consequence of tumour invasion and the appearance of metastasis, with an emphasis on metastasis modulated from extrinsic factors of the tumour environment and how the first line of defence could promote cancer-associated inflammation or participate in the metastatic cascade. Subsequently, they describe the different types of cells, grouping into three groups according to the authors’ criteria.

The authors make an extensive review of a topic already developed and with multiple studies at different levels. However, the metastasis is a topic in full development at the present. Many recent and important references are used, the topics are well arranged and a very scientific and clear writing of language is used. All these things facilitate the understanding and make it easy to read. The quality of the figure is good.

Minor comments:

- Page 2, chapter 2: The authors describe different effects by which the immune system contributes to the removal of tumour cells or support the invasion and metastasis of other organs. The authors should describe more deeply the effect of cytokines secreted by immune cells on tumour progression.  Is there any reference in the literature on how the abundance of certain cells from TME influences the prognosis of cancer?

Due to the diverse effects/associations reported on different cytokines secreted by different cell types, we have discussed their clinical association within the section related to each cell type and not in Page 2, which was more a general introduction. In addition, this review focuses specifically on whether the immune cells if they were evident/involved in relevant steps of tumor invasion and metastasis, and not whether they are attributed to a favorable/unfavorable prognosis. 

- Page 5: It would be interesting if the authors could broaden the topic of the relationship between neutrophils-MDSCs-NK cells

Whilst there is an extensive study around the relationship between NK cells and neutrophils / M and GMDSCs, majority of which were focused on the immune suppressive influence of these cells on NK cell anti-tumour activity. However, how these cells were involved specifically during the metastatic process, which is the focus of this review, is relatively limited.

- Page 7: The immunosuppressive and protective properties of ILC3 and its contribution to the development of TLSs in lung cancer should be describe more deeply as it is reported in the reference 148.

We have included the contribution of NCR-expressing ILC3s in Page. 7. Reference 148 and 149 were swapped for the content.

The increased density of ILC3s was reported to be associated with increased lymphatic tumor cell invasion and metastasis in human breast cancer patients [148] and with a subset of NCR-expressing ILC3s which were proposed to contribute to intratumoral lymphoid structures in human lung cancer [149]. These NCR+ILC3s were mainly localized at the edge of tumor-associated tertiary lymphoid structures, to produce IL-22, TNF-a, IL-8 and IL-2, and when activated are able to activate endothelial cells [149].

Reviewer 4 Report

This was a very nice review article that summarizes the role of the innate immune system in tumor immunity and metastasis. They start by briefly describing the metastatic cascade, then summarize studies showing roles for each innate immune cell type in cancer progression and metastasis. The article is well-written and appears thorough. It is very timely given the emergence of immune therapies as effective treatments in a number of cancers. I believe this will be of interest to a broad audience.

I have only 3 minor suggestions:

  • The authors nicely summarize roles for each innate immune cell type, and they also include figure 2 which provides a visual summary of what they discuss. However, I wonder if the article would benefit from a table that lists each immune cell type and its function(s) in cancer. Collectively these cell types have both pro and anti-tumor and metastasis functions, and in some cases the same cell type can promote or inhibit cancer progression. A table that allows the reader to quickly see how each cell type influences cancer would help us see that larger picture. Other reviews on the immune system and tumor stroma have included such tables and they are very effective. The authors could include the same references they list in the text in the table.
  • The authors suggest that extravasation is the final step of the metastatic cascade, really the final step is survival and outgrowth of the disseminated tumor cell into a detectable metastasis. They mention this in section 1 and figure 1, but the text says “Extravasation is the final step”. They should just change their text to not mislead the reader.
  • There are several minor grammatical errors and typos that should be corrected.

Author Response

This was a very nice review article that summarizes the role of the innate immune system in tumor immunity and metastasis. They start by briefly describing the metastatic cascade, then summarize studies showing roles for each innate immune cell type in cancer progression and metastasis. The article is well-written and appears thorough. It is very timely given the emergence of immune therapies as effective treatments in a number of cancers. I believe this will be of interest to a broad audience.

I have only 3 minor suggestions:

  • The authors nicely summarize roles for each innate immune cell type, and they also include figure 2 which provides a visual summary of what they discuss. However, I wonder if the article would benefit from a table that lists each immune cell type and its function(s) in cancer. Collectively these cell types have both pro and anti-tumor and metastasis functions, and in some cases the same cell type can promote or inhibit cancer progression. A table that allows the reader to quickly see how each cell type influences cancer would help us see that larger picture. Other reviews on the immune system and tumor stroma have included such tables and they are very effective. The authors could include the same references they list in the text in the table.

A Table (Table 1) summarising the potential role of immune cells in cancer and in metastatic cascade is included in the manucript.

  • The authors suggest that extravasation is the final step of the metastatic cascade, really the final step is survival and outgrowth of the disseminated tumor cell into a detectable metastasis. They mention this in section 1 and figure 1, but the text says “Extravasation is the final step”. They should just change their text to not mislead the reader.

We have amended the “final” to “following” steps.

  • There are several minor grammatical errors and typos that should be corrected.

We have proofread the manuscript for errors.

Round 2

Reviewer 1 Report

The authors have partially response a the integration of review.

Author Response

Thank you.